# Defect of Interferon γ Leads to Impaired Wound Healing through Prolonged Neutrophilic Inflammatory Response and Enhanced MMP-2 Activation

**DOI:** 10.3390/ijms20225657

**Published:** 2019-11-12

**Authors:** Emi Kanno, Hiromasa Tanno, Airi Masaki, Ayako Sasaki, Noriko Sato, Maiko Goto, Mayu Shisai, Kenji Yamaguchi, Naoyuki Takagi, Miki Shoji, Yuki Kitai, Ko Sato, Jun Kasamatsu, Keiko Ishii, Tomomitsu Miyasaka, Kaori Kawakami, Yoshimichi Imai, Yoichiro Iwakura, Ryoko Maruyama, Masahiro Tachi, Kazuyoshi Kawakami

**Affiliations:** 1Department of Science of Nursing Practice, Tohoku University Graduate School of Medicine, 2-1 Seiryo-cho, Aoba-ku, Sendai 9808575, Japan; hiromasa-tanno@med.tohoku.ac.jp (H.T.); skn.kdr.oor.325@gmail.com (M.G.); ambystomatmexicanum@gmail.com (M.S.); maruyama@med.tohoku.ac.jp (R.M.); 2Department of Plastic and Reconstructive Surgery, Tohoku University Graduate School of Medicine, 2-1 Seiryo-cho, Aoba-ku, Sendai 9808575, Japan; masaki.a.m.825@gmail.com (A.M.); ssayakoss812@gmail.com (A.S.); surry310630@yahoo.co.jp (N.S.); takagi-prs@med.tohoku.ac.jp (N.T.); miki_shouji_0121@yahoo.co.jp (M.S.); yo-imai@med.tohoku.ac.jp (Y.I.); tachi@med.tohoku.ac.jp (M.T.); 3Department of Medical Microbiology, Mycology and Immunology, Tohoku University Graduate School of Medicine, 2-1 Seiryo-cho, Aoba-ku, Sendai 9808575, Japanyuki.m4058@gmail.com (Y.K.); ishii-k@med.tohoku.ac.jp (K.I.); 4Department of Intelligent Network for Infection Control, Tohoku University Graduate School of Medicine, 2-1 Seiryo-cho, Aoba-ku, Sendai 9808575, Japan; ko-sato@med.tohoku.ac.jp (K.S.);; 5Division of Pathophysiology, Department of Pharmaceutical Sciences, Faculty of Pharmaceutical Sciences, Tohoku Medical and Pharmaceutical University, Sendai 9818558, Japan; t-miya13@tohoku-mpu.ac.jp (T.M.); fl.nikomako.dr@gmail.com (K.K.); 6Division of Laboratory Animals, Research Institute for Biomedical Sciences, Tokyo University of Science, 2669 Yamazaki, Noda, Chiba 2788510, Japan; iwakura@rs.tus.ac.jp

**Keywords:** interferon-γ, wound healing, neutrophils, matrix metalloproteinase-2

## Abstract

Interferon (IFN)-γ is mainly secreted by CD4+ T helper 1 (Th1), natural killer (NK) and NKT cells after skin injury. Although IFN-γ is well known regarding its inhibitory effects on collagen synthesis by fibroblasts in vitro, information is limited regarding its role in wound healing in vivo. In the present study, we analyzed how the defect of IFN-γ affects wound healing. Full-thickness wounds were created on the backs of wild type (WT) C57BL/6 and IFN-γ-deficient (KO) mice. We analyzed the percent wound closure, wound breaking strength, accumulation of leukocytes, and expression levels of *COL1A1*, *COL3A1*, and matrix metalloproteinases (MMPs). IFN-γKO mice exhibited significant attenuation in wound closure on Day 10 and wound breaking strength on Day 14 after wound creation, characteristics that are associated with prolonged neutrophil accumulation. Expression levels of *COL1A1* and *COL3A1* mRNA were lower in IFN-γKO than in WT mice, whereas expression levels of *MMP-2* (gelatinase) mRNA were significantly greater in IFN-γKO than in WT mice. Moreover, under neutropenic conditions created with anti-Gr-1 monoclonal antibodies, wound closure in IFN-γKO mice was recovered through low *MMP-2* expression levels. These results suggest that IFN-γ may be involved in the proliferation and maturation stages of wound healing through the regulation of neutrophilic inflammatory responses.

## 1. Introduction

Wound healing is a complex process involving inflammation, cell proliferation, matrix deposition, and tissue remodeling [1,2]. During the inflammatory phase, infiltrating neutrophils and macrophages play an important role in the defense against bacterial infections and debridement of necrotic tissue [2]. In the proliferation phase, fibroblasts and myofibroblasts interact and produce extracellular matrix (mainly collagen), resulting in granulation tissue formation. The tissue remodeling process is associated with tissue maturation and collagen degradation by matrix metalloproteinases (MMPs), which are mainly derived from leukocytes and dermal fibroblasts [3]. Previously, several MMPs including MMP-2, -8, -9 and -13 have been reported to be involved in wound healing [4,5,6,7].

Neutrophils are the first infiltrating cells to appear within 24 h after wound creation and are necessary for host defense responses [8,9]. However, prolonged neutrophil infiltration is involved in the degradation of collagen by the production of proteinases such as MMPs. In-vitro collagen synthesis by fibroblasts is induced by transforming growth factor-β (TGF-β) [10,11] and inhibited by interferon (IFN)-γ [12,13].

IFN-γ is mainly secreted by CD4+ helper T cells, NK cells, and NKT cells and contributes to the activation of immune cells [14]. IFN-γ is also associated with both neutrophil recruitment and cell clearance through apoptosis [15]. Indeed, in the thrombus resolution process, which resembles wound healing, the absence of IFN-γ accelerates thrombus resolution by enhancing MMP-9 but not MMP-2 [16]. As for IFN-γ’s role in wound healing, in a mouse acute open wound model [14] and a post-scald burn injury model [17], IFN-γKO mice exhibited accelerated healing and enhanced TGF-β expression compared with WT mice, suggesting that IFN-γ makes a negative contribution to the skin wound healing process. While treatment with TNF-α plus IFN-γ-stimulated monocytes/macrophages in diabetic rat wounds improved the delay in wound healing [18], IFN-γ’s role in wound healing remains controversial.

With this background, we focused on the effects of IFN-γ deficiency on the proliferation phase of skin wound healing using a mouse model with full-thickness wounds. Here, we show that IFN-γ is required for the repair of skin wounds in the proliferation phase due to its regulation of neutrophilic inflammatory responses, including the activation of MMP-2 (Gelatinase A) which is mainly derived from neutrophils.

## 2. Results

### 2.1. Delayed Wound Healing in IFN-γ-Deficient Mice in the Proliferative Phase

To examine the possible contribution of IFN-γ to wound healing, the rate of wound closure in IFN-γKO mice was compared with that in WT mice. Wound closure on Day 10 was significantly delayed in IFN-γKO mice compared with WT mice (Figure 1A,B). To confirm this effect, wound breaking strength was examined. Wound breaking strength on Day 14 was significantly delayed in IFN-γKO mice compared with WT mice (Figure 1C). As an alternate indicator of wound healing, we also evaluated α-SMA, which indicates myofibroblast differentiation. As shown in Figure 1D, the number of α-SMA^+^ cells was significantly decreased in IFN-γKO mice. In addition, IFN-γKO mice exhibited lower *COL1A1*, *COL3A1*, and *TGF-β1* expression compared with WT mice on Day 14 (Figure 1E).

### 2.2. Prolonged Accumulation of Neutrophils in IFN-γKO Mice

To define the role of inflammatory leukocytes during the wound healing process in IFN-γKO mice, wounded skin tissues were histologically examined in IFN-γKO and WT mice. As shown in Figure 2A, the former genotype exhibited prolonged accumulation of inflammatory leukocytes at the wound sites on Day 7. In the WT mice, in contrast, mainly fibroblasts were accumulated at the wound sites. Next, Ly6G, a marker specific to neutrophils, given that accumulated macrophages and eosinophils at the wound sites did not express Ly6G [19], was evaluated histologically. As shown in Figure 2B, the number of Ly6G^+^ cells on Day 7 was significantly greater in IFN-γKO mice. Consistent with these results, *CXCL1* (KC) and *CXCL2* (MIP-2) expression levels were also significantly higher in IFN-γKO mice than in WT mice on Day 7 (Figure 2C).

### 2.3. Inhibited MMP-2 Activation by IFN-γ

To define the mechanisms underlying IFN-γ-associated reductions in breaking strength and in *COL1A1* and *COL3A1* expression as well as IFN-γ-associated prolonged neutrophil accumulation, we examined mRNA expression levels of the collagen degradation-associated factors *MMP-2* and *MMP-9* in the wounded tissue. *MMP-2* mRNA expression on Day 14 was significantly increased in IFN-γKO mice compared with WT mice; with regard to *MMP-9* expression, in contrast, there was no significant difference between WT and IFN-γKO mice (Figure 3A). As shown in Figure 3B, from a morphological perspective, *MMP-2* is mainly expressed in neutrophils in IFN-γKO mice in contrast to WT mice. Next, because *MMP-2* expression was significantly increased in IFN-γKO mice, we examined the involvement of IFN-γ in the activity of neutrophil-derived MMP-2 and pro-MMP-2 activity by gelatin zymography. As shown in Figure 3C,D, pro-MMP-2 activity level was significantly suppressed by IFN-γ in a concentration-dependent manner, while MMP-2 activity, in contrast, was not detected in any experimental groups.

### 2.4. Wound Healing and MMP-2 Expression after Neutrophil Depletion Induced by Anti-Gr-1 Monoclonal Antibody in IFN-γKO Mice

As histological findings have revealed, MMP-2 derived mainly from neutrophils is involved in the delayed wound healing in IFN-γKO mice, as described above. Accordingly, we examined the effect of neutropenia induced by means of the anti-Gr-1 monoclonal antibody on wound closure and *MMP-2* expression. As shown in our recent study [20], the neutrophils in peripheral blood are completely depleted by this treatment. Wound closure on Day 10 was significantly accelerated in anti-Gr-1 antibody-treated mice compared with control IgG-treated mice (Figure 4A). As shown in Figure 4B,C, the accumulation of Ly6G^+^ neutrophils had almost completely disappeared on Day 10 after anti-Gr-1 antibody administration at the wound sites. In addition, *MMP-2* expression was significantly decreased in anti-Gr-1 antibody-treated mice (Figure 4D). In the control group, interestingly, *MMP-2* was mainly detected in infiltrating leukocytes, whereas *MMP-2*-expressing fibroblasts were frequently detected in the anti-Gr-1 antibody-treated group (Figure 4E).

## 3. Discussion

In the current study, IFN-γKO mice exhibited significant attenuation in wound closure, wound breaking strength, and myofibroblast differentiation in the proliferation phase compared with WT mice through prolonged neutrophil accumulation and enhanced MMP-2 activation. 

IFN-γ contributes to macrophage activation [14], neutrophil recruitment, and cell clearance by apoptosis [15]. Yet the question of how IFN-γ contributes to wound healing, especially in the proliferative phase, remains controversial and poorly understood. Regarding the role of IFN-γ in skin wounds, we previously reported that IFN-γ plays a key role in the early phase of the wound healing process in a study on mice deficient in invariant natural killer T (*i*NKT) cells, which are major IFN-γ-producing cells [21]. In addition, IFN-γ-treated LEPCs, which initiate blood vessel regeneration [22], or TNF-α- and IFN-γ-treated monocytes/macrophages [18] have been reported to be involved in the promotion of wound healing. In the current study, IFN-γKO mice exhibited significant attenuation in wound closure on Day 10 in association with prolonged neutrophil accumulation. In contrast to this, Ishida and Kondo et al. [14] demonstrated that IFN-γ deficiency accelerated the wound healing process in association with an early-phase reduction in the infiltration of myeloperoxidase (MPO)^+^ neutrophils, F4/80^+^ macrophages, and CD3^+^ T cells. 

Our murine wound model was at low risk for microbial infection as we used a clean procedure for the wounding and occlusive dressings for the wounds (closed wounds) until tissue collection. The model used by Ishida and Kondo et al. [14], in contrast, analyzed open wounds, i.e., wounds that had not been covered with occlusive dressings. This difference may have affected the different results of our two studies as a variation in environmental moisture and the microbial load at the wound site may have affected the findings. This possibility is strengthened by our previous finding that, compared with WT mice, IL-17AKO mice exhibited accelerated wound healing under closed-wound conditions but delayed wound healing under open-wound conditions [23].

In the present study, IFN-γKO mice exhibited diminished wound breaking strength, reduced myofibroblast differentiation, and low levels of *COL1A1* and *COL3A1* expression. Although several reports have demonstrated that IFN-γ can inhibit collagen synthesis by fibroblasts in vitro [12,13], in wound sites, IFN-γ can contribute to collagen deposition [21]. Previously, Hata et al. reported that TGF-β1 induces myofibroblast differentiation and collagen synthesis in the proliferation phase [24]. In this study, TGF-β1 expression was decreased in IFN-γKO mice compared with WT mice. Thus, our results are likely to be related to a delay in collagen synthesis.

In the current study, IFN-γKO mice exhibited delayed wound repair in the proliferative phase along with an increased neutrophil count, suggesting that accumulated neutrophils may suppress the healing process. In normal acute wounds, neutrophils are infiltrated immediately after skin injury and initially play a key role in antimicrobial activity; later, these cells undergo apoptosis and are engulfed by macrophages [25]. In non-healing wounds, however, prolonged neutrophil accumulation often leads to persistent inflammation through the production of proteases such as MMPs [26]. The functions of *MMP-2* [6], *MMP-8* [5], and *MMP-9* [27] have been studied with regard to the wound healing process. MMP-2 is not considered to play a critical role in normal acute murine wounds [6]. In non-healing wounds in humans, however, high levels of *MMP-2* activity have been detected [28]. In the current study, IFN-γKO mice exhibited delayed wound healing associated with a significant increase in *MMP-2* expression on recruited neutrophils at the wound sites. We also confirmed that pro-*MMP-2* activity levels in peritoneal neutrophils were significantly suppressed by IFN-γ stimulation. In fact, our current results demonstrate that delayed wound healing in IFN-γKO mice can be recovered under neutropenic conditions induced by treatment with the anti-Gr-1 monoclonal antibody, and that this recovery is associated with low levels of MMP-2. Previously, Qin et al. [29] reported the transcriptional suppression of *MMP-2* gene expression in human astroglioma cells by IFN-γ administration. Our skin wound model likewise suggests that IFN-γ could be involved in *MMP-2* expression.

MMP-2 has been reported to involve tissue remodeling by degrading extracellular matrix components such as type III collagen, type IV collagen, fibronectin and elastin [30,31]. At wound sites, upregulation of *MMP-2* expression has been observed in both granulation and scar tissues after skin injury [32]: during normal wound repair, *MMP-2* expression reached peak levels on Day 3 after wound creation and declined thereafter to the baseline level [33]. In this study, we showed that IFN-γKO mice exhibited decreased wound breaking strength along with upregulated *MMP-2* activity, suggesting that MMP-2 in the proliferative phase may reduce the strength of wounded skin. Thus, MMP-2 is likely to contribute positively in the early phase of wound healing, and negatively from the proliferative phase onward.

In this study, our in vitro gelatin zymography experiment detected pro-*MMP-2* activity but not MMP-2 itself in peritoneal neutrophils. It has previously been reported that MMP-2 is secreted as a zymogen (pro-MMP-2), and that membrane-bound MMP-14 (MT-1-MMP), which is mainly expressed on fibroblasts and cancer cells, activates secreted MMP-2 by cleaving its pro-domain [34]. Thus, the absence of MMP-14-expressing cells such as fibroblasts may be related to the absence of detectable MMP-2 in our in vitro analysis.

In conclusion, the present study demonstrated that IFN-γ plays an important role in the proliferation phase of skin wound healing and in the neutrophilic inflammatory response at the wound site. To date, little was known about IFN-γ’s function in the proliferation phase; here, we have shown that it contributes significantly to wound strength and the suppression of inflammation. Inflammation is deeply involved in wound healing [35], but little is known about therapy for inflammatory responses at the wound sites. IFN-γ therapy has already been used as a treatment for pulmonary fibrosis [36] and may also be useful in a novel approach to the treatment of augmented fibrosis in the skin, such as hypertrophic scarring, keloid scarring and scleroderma. However, we did not confirm the effects of IFN-γ administration in this study. Further investigation is necessary to clarify the effects of IFN-γ therapy on augmented fibrosis in skin as well as its optimal dose and route.

## 4. Materials and Methods 

### 4.1. Animals

IFN-γ gene-disrupted (knockout (KO)) mice were generated and established as described previously [37] and backcrossed to C57BL/6 mice for more than eight generations. Wild-type (WT) C57BL/6 mice, purchased from CLEA Japan (Tokyo, Japan), were used as controls. Male or female mice at 7 to 10 weeks of age were used in the experiments. Food and water were available ad libitum. All mice were kept under specific pathogen-free conditions in the Institute for Animal Experimentation, Tohoku University Graduate School of Medicine (Sendai, Japan). All experimental protocols described in the present study were approved by the Ethics Review Committee for Animal Experimentation of Tohoku University (2016MdA-279-3, 19 July 2016; 2016MdLMo-138-3, 7 July 2016). All experiments were performed under anesthesia, and all efforts were made to minimize suffering of the animals.

### 4.2. Wound Creation and Tissue Collection

All handling of the animals was performed under anesthesia induced by an intraperitoneal injection of 40 mg/kg sodium pentobarbital (Somnopentyl, Kyoritsu Seiyaku Corporation, Tokyo, Japan) and sustained by inhalation anesthesia of isoflurane (Isoflurane, Mairan Pharma, Osaka, Japan). The dorsal hair was shaved to fully expose the n skin, which was then rinsed with 70% ethanol. Four full-thickness wounds extending to the panniculus carnosus were created using a 6 mm diameter biopsy punch (Biopsy Punch, Kai industries Co., Ltd., Gifu, Japan) under sterile conditions. The injured areas were covered with a polyurethane film (Tegaderm Transparent Dressing, 3M Health Care, St. Paul, MN, USA) and an elastic adhesive bandage (Hilate, Iwatsuki, Tokyo, Japan) as an occlusive dressing. The day on which the wounds were made was designated as Day 0. At various time points, mice were sacrificed, and the wound tissue was collected by excising a 1 cm square section of skin using scissors and a surgical knife.

### 4.3. Administration of Anti-Gr-1 Antibody and the Effect of Neutrophil Depletion Induced by This Means

Anti-Gr-1 monoclonal antibody was purified from hybridoma culture supernatants (clones RB6-8C5) using a protein G column kit (Kirkegaard & Perry Laboratories, Gaithersburg, MD, USA). To neutralize the biological activity of neutrophils, mice were injected intraperitoneally with 400 µg of mAb on Days 5 and 7 after wounding. Rat IgG (ICN Pharmaceuticals, Aurora, OH, USA) was used as a control antibody. Immediately prior to injection and on Days 1, 2 and 5 after injection, mouse blood was collected via the tail vein and reacted with 0.83% ammonium chloride and Tris-HCl (pH 7.2), then washed three times with 1% FCS RPMI 1640 medium, yielding the blood cells used in our flow cytometric analysis. These blood cells were stained with PE-CD11b (BioLegend, San Diego, CA, USA) and APC/Cy7-anti-Ly6G mAb (clone 1A8; BioLegend). Isotype-matched irrelevant IgG was used for control staining. The stained cells were analyzed using a BD FACS Canto II flow cytometer (BD Bioscience, San Jose, CA, USA).

### 4.4. Measurement of the Wound Area

Morphometric analysis was performed on digital images obtained using a digital camera (CX4; Ricoh, Tokyo, Japan). After the wounds were created, photographs were taken of each wound before dressing. At various time points, the polyurethane films were gently removed from the experimental mice, and the wounds were photographed. Each wound area was quantified by tracing its margin and calculating the pixel area using AxioVision imaging software Release 4.6 (Carl Zeiss Micro Imaging Japan, Tokyo, Japan). Percentage of wound closure was calculated using the following formula: % wound closure = (1 − wound area at the indicated time point/wound area on Day 0) × 100.

### 4.5. Wound Breaking Strength

Wounded skin tissue was harvested from WT and *IFN-γ*KO mice on Day 14 after wound creation. A strip of this tissue located 5 mm from the center of the wound was excised with a no. 15 surgical blade (Feather Safety Razor Co., Ltd., Osaka, Japan). Wound breaking strength was measured using an IMS-001 (Keisei Medical Industrial Co., Ltd., Tokyo, Japan) as previously described [38]. Briefly, each side of the strip was pinched in a clip and the two clips were pulled apart at a constant speed of 3 cm/min until the strip broke. The result was expressed as the tensile force necessary to break the repaired wounds. Tensile force is characterized as tissue fragility and delay of collagen synthesis.

### 4.6. Histology and Immunohistochemistry

The tissues were fixed with 4% paraformaldehyde-phosphate buffer solution and embedded in paraffin. Sections were taken from the central portion of the wound and stained with hematoxylin-eosin (HE) according to the standard method. 

For immunohistochemical analysis, after endogenous peroxidase was blocked with methanol/hydrogen peroxide, the sections were incubated with 10% normal rabbit serum for 20 min to block non-specific binding and then stained with anti-α-smooth muscle actin (α-SMA) antibody (dilution 1:200; Vector Laboratories, Inc., Burlingame, CA, USA), anti-Ly6G Ab (clone 1A8; dilution 1:100; BioLegend), or anti-MMP-2 (dilution 1:200; Chemicon, Darmstadt, Germany). The sections were incubated with peroxidase-conjugated secondary Ab (4 µg/mL; Histofine Simple Stain MAX-PO, Nichirei Bioscience, Tokyo, Japan), then reacted with 3, 3-diaminobenzidine (DAB) (Nichirei Bioscience) or Alkaline Phosphatase (Dako, Bettingen, Switzerland). The number of myofibroblasts and neutrophils in six random fields (each 0.2 mm^2^) was determined by counting the number of α-SMA-positive cells or the number of Ly6G-positive cells, respectively. All analyses were performed under blinded conditions. 

### 4.7. RNA Extraction and Quantitative Real-Time RT-PCR

Total RNA was extracted from the wound tissues using ISOGEN (Nippon Gene Co. Ltd., Tokyo, Japan), and first-strand cDNA was synthesized using the PrimeScript first-strand cDNA synthesis kit (TaKaRa Bio Inc., Otsu, Japan) according to the manufacturer’s instructions. Quantitative real-time PCR was performed in a volume of 20 µL using gene-specific primers and FastStart essential DNA green master mix (Roche Applied Science, Penzburg, Germany) in a Step One^TM^ (Thermo Fisher Scientific, Waltham, MA, USA). Primers were as follows: 5′- TGT TCA GCT TTG ACC TCC G -3′ (Forward) and 5′- TAC CTC GGG TTT CCA CGT CTC A -3′ (Reverse) for COL1A1, 5′- GGA CCA GGC AAT GAT GGA AAA CC -3′ (Forward) and 5′- ACC AGG GAA ACC CAT GAC ACC -3′ (Reverse) for COL3A1, 5′-CCG CGC CTA TCG CCA ATG AGC TGC GC-3′ (Forward) and 5′-CTT GGG GAC ACC TTT TAG CAT CTT TTG G-3′ (Reverse) for CXCL1 (KC), 5′-CTG AAC AAA GGC AAG GCT AAC TG -3′ (Forward) and 5′-CAC ATC AGG TAC GAT CCA GGC TT -3′ (Reverse) for CXCL2 (MIP-2), 5′- CCC CTG ATG TCC AGC AAG TAG A -3′ (Forward) and 5′- AGT CTG CGA TGA GCT TAG GGA AA-3′ (Reverse) for MMP-2, 5′- CCC TGG AAC TCA CAC GAC ATC TTC-3′ (Forward) and 5′- GGT CCA CCT TGT TCA CCT CAT TTT -3′ (Reverse) for MMP-9 and 5′-GCT TCC TCC TCA GAC CGC TT-3′ (Forward) and 5′-TCG CTA ATC ACG ACG CTG GG-3′ (Reverse) for β-actin (ACTB). The reaction efficiency with each primer set was determined using standard amplifications. Target gene expression levels and that of ACTB as a reference gene were calculated for each sample using the reaction efficiency. The results were analyzed using a relative quantification procedure and are presented as expression levels relative to that of ACTB.

### 4.8. Isolation of Peritoneal Neutrophils

Thioglycolate-elicited peritoneal neutrophils were obtained from WT mice by a previously described method [39]. WT mice were intraperitoneally injected with 1.5 mL of sterile 4% thioglycolate; 12–15 h later, peritoneal lavages were performed with 10 mL PBS in each mouse. The proportion of neutrophils in the lavages was over 90% as assessed by FACS analysis of anti-CD45, anti-CD11b, and anti-Ly6G expression. Peritoneal neutrophils were recovered by centrifugation and suspended in RPMI 1640 medium supplemented with 10% FCS, 100 U/mL penicillin G, 100 µg/mL streptomycin, and 50 µM 2-mercaptoethanol. The obtained cells were cultured at 1 × 10^6^/mL with various doses of IFN-γ or lipopolysaccharide (LPS) (Sigma-Aldrich, St. Louis, MO, USA) for 24 h at 37 °C.

### 4.9. Pro-MMP-2 Assay

Pro-MMP-2 activation in the culture supernatants, peritoneal neutrophils and IFN-γ or LPS were measured using a gelatin zymography kit (Cosmo Bio, Tokyo, Japan) according to the manufacturer’s instructions. The gelatin zymography products were electrophoresed on 2% agarose gels, stained with 0.5 mg/mL ethidium bromide and observed with a ultraviolet transilluminator. Images were analyzed using Image J version 1.51 (National Institutes of Health, MD, USA).

### 4.10. Statistical Analysis

Data are expressed as the mean ± standard deviation (SD). Data analysis was performed using Welch’s *t*-test. A *p* value less than 0.05 was considered to indicate significance.

## Figures and Tables

**Figure 1 ijms-20-05657-f001:**
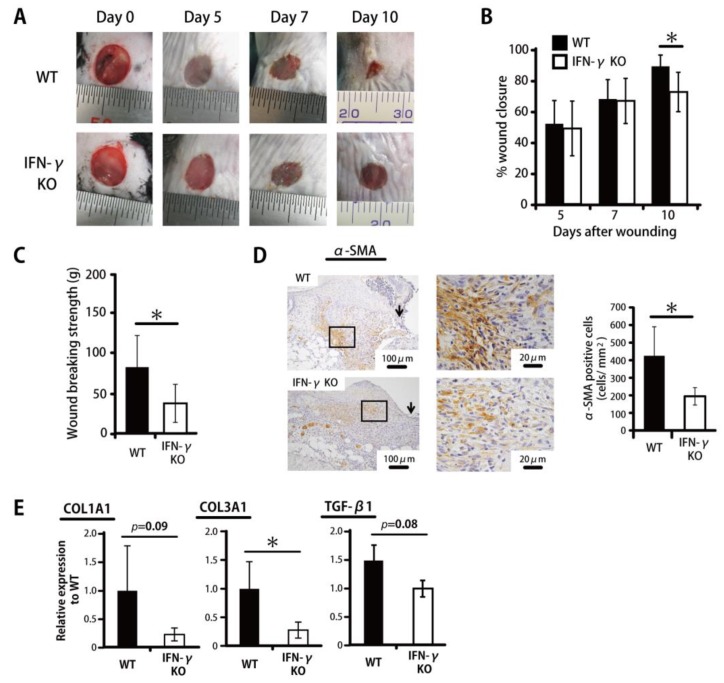
IFN-γ deficiency leads to impaired wound healing in skin. Wounds were created on the backs of WT or IFN-γKO mice. (**A**) Wound photographs in WT or IFN-γKO mice. (**B**) Percentage of wound closure was evaluated on Days 5, 7, and 10. (**C**) Wound breaking strength was measured on day 14. (**D**) The number of myofibroblasts stained with anti-α-SMA antibody on Day 10. The myofibroblast density/mm2 was determined by counting the positive cells within six visual fields (*n* = 6). Arrows indicate the re-epithelialized leading edges. (**E**) Real-time PCR was performed to detect *COL1A1*, *COL3A1*, and *TGF-β* mRNA isolated from the wound. Each column represents the mean ± SD. * *p* < 0.05.

**Figure 2 ijms-20-05657-f002:**
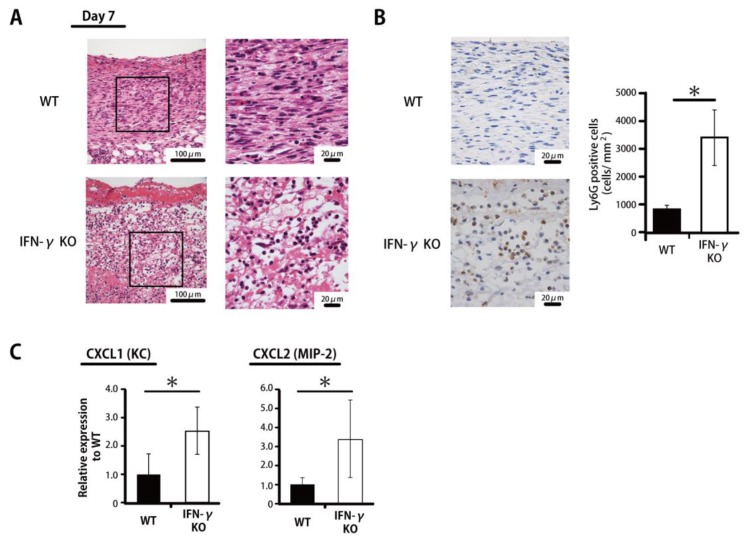
Prolonged accumulation of neutrophils in IFN-γ-KO mice. (**A**) Representative histological views of skin wounds on Day 7 are shown. (**B**) The number of neutrophils stained with anti-Ly6G antibody on Day 7. The Ly6G+ cell density/mm2 was determined by counting the positive cells in six visual fields (*n* = 6). (**C**) Real-time PCR was performed to detect *CXCL1* (KC) and *CXCL2* (MIP-2) mRNA isolated from the wound. Each column represents the mean ± SD. * *p* < 0.05.

**Figure 3 ijms-20-05657-f003:**
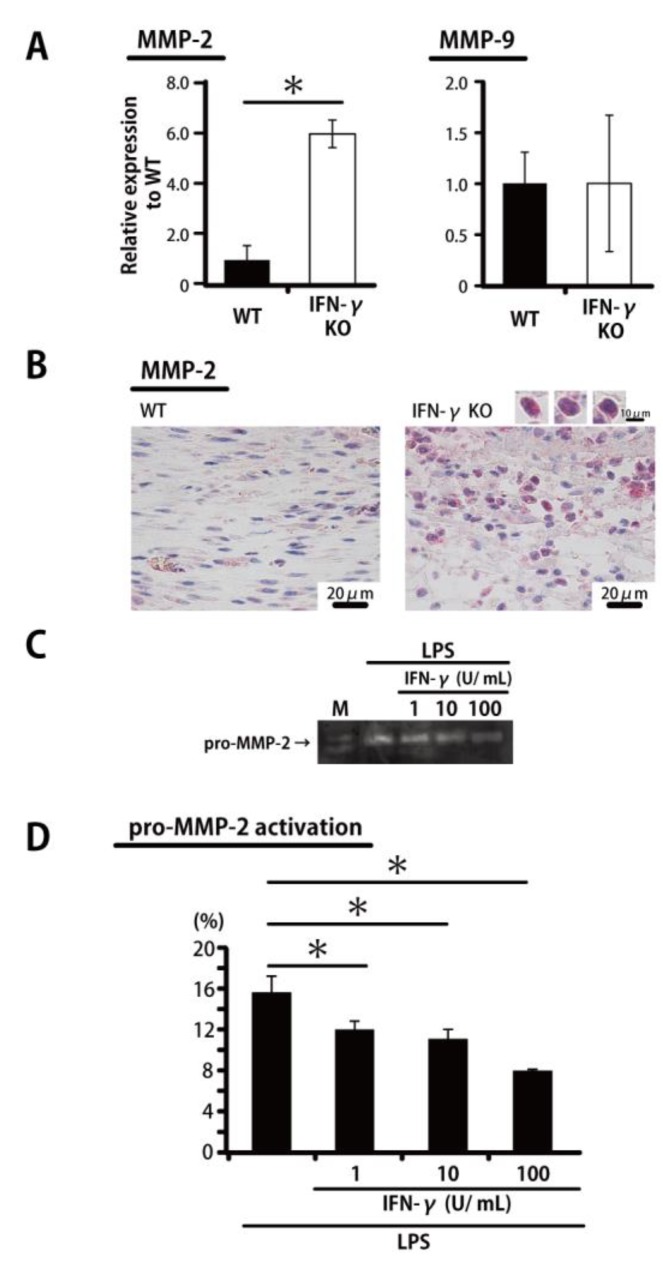
IFN-γ leads to inhibited MMP-2 activation. (**A**) Real-time PCR was performed to detect *MMP-2* and *MMP-9* mRNA isolated from the wound. (**B**) Representative histological views of wounded skin stained with MMP-2 antibody on Day 7. Red indicates MMP-2 positive cells. (**C**) Thioglycolate-elicited peritoneal neutrophils were treated with IFN-γ and lipopolysaccharide (LPS) for 24 h. The conditioned medium samples were analyzed for pro-MMP-2 activation by gelatin zymography. (**D**) The levels of pro-MMP-2 activation in (C) were analyzed using Image J image analysis software. Each column represents the mean ± SD. * *p* < 0.05. M—marker.

**Figure 4 ijms-20-05657-f004:**
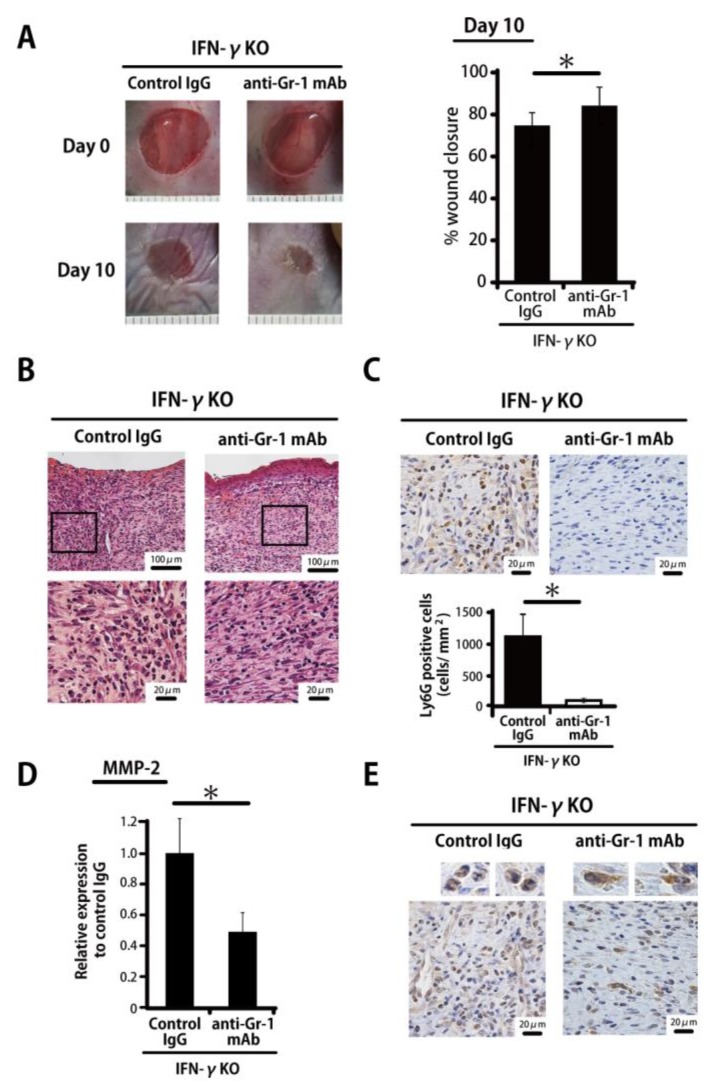
Neutrophil depletion by means of anti-Gr-1 monoclonal antibody leads to decreased MMP-2. (**A**) IFN-γKO mice were injected intraperitoneally with anti-Gr-1 monoclonal antibody or control rat IgG 5 and 7 days after wound creation. Percentage of wound closure was evaluated on Day 10. (**B**) Representative histological views of skin wounds on Day 10 are shown. (**C**) The number of neutrophils stained with anti-Ly6G antibody on Day 10. The Ly6G+ cell density/mm^2^ was determined by counting the positive cells in six visual fields (*n* = 6). (**D**) Real-time PCR was performed to detect *MMP-2* mRNA isolated from the wound. (**E**) Representative histological views of wounded skin stained with MMP-2 antibody on Day 10. Each column represents the mean ± SD. * *p* < 0.05.

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
