# Peer review of "Defect of Interferon γ Leads to Impaired Wound Healing through Prolonged Neutrophilic Inflammatory Response and Enhanced MMP-2 Activation"

_ijms, 2019, doi:10.3390/ijms20225657_

Round 1

Reviewer 1 Report

The study of Kanno et al. describes the role of IFN-γ in wound healing in vivo. The study analyzed how defect of IFN-γ affected wound healing. Full-thickness wounds were created on the backs of wild type (WT) C57BL/6 and IFN-γ-deficient (KO) mice. IFN-γKO mice exhibited significant attenuation in wound closure on day 10. Expression levels of COL1A1 and COL3A1 mRNA were lower in IFN-γKO than in WT mice. The authors suggested that IFN-γ may be involved in the proliferation and maturation stages of wound healing through the regulation of neutrophilic inflammatory responses.

Although the paper is clearly presented, some major points have to be addressed in more details:

Major revisions

1) What is exactly the wound breaking strength? it is difficult to understand.

2) Ly6G is a general marker for myeloid cells; it stains monocytes, granulocytes, and neutrophils. It is not a neutrophil specific marker. The authors should use some additional markers to clearly define neutrophils. With Ly6G alone is does not work.

3) According to my comment #2, these statements have to be confirmed by additional experiments:

“(Figure 3A). As shown in Figure 3B, 124 MMP-2-expressing cells such as neutrophils were found to be markedly increased in IFN-γKO mice 125 compared with WT mice. Next, because MMP-2 expression was significantly increased in IFN-γKO 126 mice, we examined the involvement of IFN-γ in the activity of neutrophil-derived MMP-2 and 127 pro-MMP-2 activity by gelatin zymography.”

“Because MMP-2 derived from neutrophils is involved in the delayed wound healing in 141 IFN-γKO mice as described above, we examined the effect of neutropenia induced by means of 142 anti-Gr-1 monoclonal antibody on wound closure and MMP-2 expression.”

4) It is known that MMP-2 is ALSO expressed by fibroblasts and myofibroblasts, therefore this statement has to be validated:

“Because MMP-2 derived from neutrophils is involved in the delayed wound healing in 141 IFN-γKO mice as described above, we examined the effect of neutropenia induced by means of (…)”

Author Response

Ms No. ijms-615510, titled “Defect of interferon γ leads to impaired wound healing through prolonged neutrophilic inflammatory response and enhanced MMP-2 activation” by Emi Kanno et al.

Reply to Reviewer #1:

The authors are grateful to the reviewer for these positive and constructive comments, which were very helpful in improving our paper. We have replied point-by-point to the comments, as shown below, and have taken all of the comments into consideration in revising our manuscript. The revisions are shown in red font with yellow highlighting.

The study of Kanno et al. describes the role of IFN-γ in wound healing in vivo. The study analyzed how defect of IFN-γ affected wound healing. Full-thickness wounds were created on the backs of wild type (WT) C57BL/6 and IFN-γ-deficient (KO) mice. IFN-γKO mice exhibited significant attenuation in wound closure on day 10. Expression levels of COL1A1 and COL3A1 mRNA were lower in IFN-γKO than in WT mice. The authors suggested that IFN-γ may be involved in the proliferation and maturation stages of wound healing through the regulation of neutrophilic inflammatory responses.

Although the paper is clearly presented, some major points have to be addressed in more details:   

Major Points:

1) What is exactly the wound breaking strength? it is difficult to understand.

Reply: We appreciate this important comment. In our previous study (Tanno H, et al., Am J Pathol. 185: 3248-3257, 2015; Kanno E et al., Exp Dermatol., 26: 1097-1104, 2017) we described wound breaking strength in detail. We now cite this paper as a reference and have added some explanation of ‘wound breaking strength’ in the Materials and Methods section. In keeping with this alteration, we also added some other text in the Materials and Methods section (page 9, lines 294 to 295).

2) Ly6G is a general marker for myeloid cells; it stains monocytes, granulocytes, and neutrophils. It is not a neutrophil specific marker. The authors should use some additional markers to clearly define neutrophils. With Ly6G alone is does not work.

Reply: Thank you for the insightful comment. We understand that Ly6G is expressed mainly in neutrophils, although other myeloid cells including eosinophils and differentiating monocytes also express it (Percopo CM et al., J Leukoc Biol., 101: 321-328, 2017; Lee PY et al., J Leukoc Biol., 94: 585-594, 2013), as per your suggestion. However, it has been reported that accumulated macrophages and eosinophils at the wound site did not express Ly6G (Daley JM et al., J Leukoc Biol., 83: 64-70, 2008). In addition, in this study as well as in our previous investigations (Kanno E et al., Exp Dermatol., 26: 1097-1104, 2017; Tanno H et al., Wound Repair Regen., 25: 805-815, 2017; Miura T et al., J Invest Dermatol., 139: 702-711, 2019), we also confirmed morphologically that leukocytes accumulated at the wound sites through histological analysis under microscopic observation with high power magnification; in contrast, few eosinophils and monocytes were observed at the wound sites. Through morphological identification, neutrophils were easily distinguished from eosinophils and monocytes. To clarify this, we cited an additional reference and added some text in the Results section (page 3, lines 107 to 108, page 4, lines 124 to 125 and page 5, line 141).

3) According to my comment #2, these statements have to be confirmed by additional experiments:

“(Figure 3A). As shown in Figure 3B, MMP-2-expressing cells such as neutrophils were found to be markedly increased in IFN-γKO mice compared with WT mice. Next, because MMP-2 expression was significantly increased in IFN-γKO mice, we examined the involvement of IFN-γ in the activity of neutrophil-derived MMP-2 and pro-MMP-2 activity by gelatin zymography.”

“Because MMP-2 derived from neutrophils is involved in the delayed wound healing in IFN-γKO mice as described above, we examined the effect of neutropenia induced by means of anti-Gr-1 monoclonal antibody on wound closure and MMP-2 expression.”

Reply: The authors thank the reviewer for this important comment. As described in our reply to your comment #2, we morphologically confirmed that these cells were neutrophils, and accordingly made the abovementioned changes to the Results section (page 4, lines 124 to 125 and page 5, line 141).

4) It is known that MMP-2 is ALSO expressed by fibroblasts and myofibroblasts, therefore this statement has to be validated:

“Because MMP-2 derived from neutrophils is involved in the delayed wound healing in IFN-γKO mice as described above, we examined the effect of neutropenia induced by means of (…)”

Reply: The authors thank the reviewer for this important comment. As described in our reply to your comment #2, we morphologically identified the cells as neutrophils and not as fibroblasts or myofibroblasts, and accordingly made the abovementioned changes to the Results section (page 4, lines 124 to 125 and page 5, line 141).

Reviewer 2 Report

MS is particularly interesting, just some corrections. in particular, I suggest to pay more attention to the conclusions section, and to modify materials and methods section in order to have more details to be reproducible.

Conclusions and last part of the MS need to be rephrased in order to put more attention to obtained results as well as to possible applications of the observations.

Author Response

Ms No. ijms-615510, titled “Defect of interferon γ leads to impaired wound healing through prolonged neutrophilic inflammatory response and enhanced MMP-2 activation” by Emi Kanno et al.

Reply to Reviewer #2:

The authors are grateful to the reviewer for these positive and constructive comments, which were very helpful in improving our paper. We have replied point-by-point to the comments, as shown below, and have taken all of the comments into consideration in revising our manuscript. The revisions are shown in red font with yellow highlighting.

MS is particularly interesting, just some corrections. in particular, I suggest to pay more attention to the conclusions section, and to modify materials and methods section in order to have more details to be reproducible.

 Conclusions and last part of the MS need to be rephrased in order to put more attention to obtained results as well as to possible applications of the observations.

Reply: We appreciate this important comment. As per your suggestion, we added some text about the possible applications to the Conclusion subsection (page 8, lines 232 to 233). In addition, we added more details to the Materials and Methods section (page 10, lines 346 to 350).

Reviewer 3 Report

Dear Authors & Editors, 

thank you very much for this nice manuscript. 

The manuscript is well structured, the scope is clearly outlined, experiments seem to be well conducted, data presentation is good. Furthermore, the findings seem to be well embedded in the field and discussed against contrasting findings. 

The data have statistical significance and the finding are relevant and not over stated.

Overall the manuscript is very well prepared, I could not find any mis-spellings or typos on first sight which clearly highlights the care that was applied to prepare the manuscript. 

The references were chosen well, all from high quality journals and good match in general.

There is little that I can add to improve the manuscript because it already is of very high quality. Maybe, it might be nice to include one or two references from the International Journal of Molecular Sciences to relate to the work that has previously been published in the same journal.

One matching reference could be https://doi.org/10.3390/ijms19102913

but there might be others that could be a better match.

Overall a very nice manuscript.   

Author Response

Ms No. ijms-615510, titled “Defect of interferon γ leads to impaired wound healing through prolonged neutrophilic inflammatory response and enhanced MMP-2 activation” by Emi Kanno et al.

Reply to Reviewer #3:

The authors are grateful to the reviewer for these positive and constructive comments, which were very helpful in improving our paper. We have replied point-by-point to the comments, as shown below, and have taken all of the comments into consideration in revising our manuscript. The revisions are shown in red font with yellow highlighting.

Dear Authors & Editors,

thank you very much for this nice manuscript.

The manuscript is well structured, the scope is clearly outlined, experiments seem to be well conducted, data presentation is good. Furthermore, the findings seem to be well embedded in the field and discussed against contrasting findings.

The data have statistical significance and the finding are relevant and not over stated.

Overall the manuscript is very well prepared, I could not find any mis-spellings or typos on first sight which clearly highlights the care that was applied to prepare the manuscript.

The references were chosen well, all from high quality journals and good match in general.

There is little that I can add to improve the manuscript because it already is of very high quality. Maybe, it might be nice to include one or two references from the International Journal of Molecular Sciences to relate to the work that has previously been published in the same journal.

One matching reference could be https://doi.org/10.3390/ijms19102913

but there might be others that could be a better match.

Overall a very nice manuscript. 

Reply: We appreciate your very favorable comments and important suggestion. As per your suggestion, we cited your recommended reference and added some text related to it in the Discussion section (page 8, lines 232 to 238).